# Learning Transferable Sub-Goals by Hypothesizing Generalizing Features

## Abstract

Transfer is a key promise of hierarchical reinforcement learning, but requires first learning transferable skills. For an agent to effectively transfer a skill it must identify features that generalize and define the skill over this subset. However, this task is under-specified from a single context as the agent has no prior knowledge of what future tasks may be introduced. Since successful transfer requires a skill to reliably achieve a sub-goal from different states, we focus our attention on ensuring sub-goals are represented in a transferable way. For each sub-goal, we train an ensemble of classifiers while explicitly incentivizing them to use minimally overlapping features. Each ensemble member represents a unique hypothesis about the transferable features of a sub-goal that the agent can use to learn a skill in previously unseen portions of the environment. Environment reward then determines which hypothesis is most transferable for the given task, based on the intuition that useful sub-goals lead to better reward maximization. We apply these reusable sub-goals to MiniGrid and Montezuma's Revenge, allowing us to learn previously defined skills in unseen parts of the state-space.

## 1 Introduction

Hierarchical reinforcement learning (HRL) (Barto & Mahadevan, 2003) is a promising approach for scaling RL to challenging, long-horizon problems. HRL methods learn temporally extended *skills* that abstract away the details of low-level action executions. The most popular HRL framework, the *options* framework (Sutton et al., 1999), models skills using three components: a set of states from which execution can begin, a policy which picks low-level actions, and a collection of states where execution ceases, which can be represented by a *sub-goal*.

To fully realize the benefits of HRL, learned options should be transferable. For example, a robot trained to open a door in a factory should be able to open doors in a user's home with minimal additional training. However, it is challenging to generalize an option from a single context to another. This is primarily because all three components of the option are conditioned on the *entire* state—which includes spurious features unnecessary for successful execution—making generalization from *one example* challenging because it is severely under-specified. However, if these components are defined over generalizable features of the state, the option can transfer to novel contexts from a single instance. The core problem here is that, in practice, an option will first be discovered in a single context (like a factory) without foreknowledge of the circumstances under which it can be applied again in the future. When defined in a single context, a sub-goal will have many confounding factors, which cannot be untangled. We further argue this point with a toy example in Section 3.2.

How can the agent learn a state representation that will result in seamless transfer? This problem can be easily tackled in supervised learning by incorporating data from different contexts or by applying class-preserving data augmentations. However, in supervised learning, the important features have been selected prior to learning, through the dataset. If we define the class *dog*, a human has previously chosen what features represent a dog and selected a dataset accordingly. If we wish to transfer from a single context, this information is missing, resulting in a set of data that could be defined accurately in multiple ways. If we have a sub-goal that results in us opening a cupboard which contains a red ball, is the sub-goal defined by opening the cupboard or finding the red ball? From this one instance, we do not have enough information to disentangle these two features. Instead, we propose that the RL agent maintain several, diverse *hypotheses* over which features of the state

might generalize to upcoming tasks, learning a set of classifiers with minimal confounding features. The agent then selects from among these hypotheses, testing them in the environment, and updates its beliefs over which members of the ensemble have learned transferable features for a given task.

We focus on transferring sub-goals, since the agent can learn an option given a sub-goal (as we argue in Section 3). We begin by assuming access to a sub-goal (potentially discovered using an off-the-shelf option discovery method), defined for a single context. This sub-goal is used to generate labeled data which acts as a training set for an ensemble of classifiers. Each ensemble member maintains a unique hypothesis of the generalizing state-features and is, individually, a new sub-goal; then, for each hypothesis, we learn a low-level policy that aims to reach the hypothesized sub-goal. Finally, a high-level policy selects among the low-level policies to maximize extrinsic reward.

We test our approach in two simulated environments: MONTEZUMASREVENGE (Bellemare et al., 2013; Machado et al., 2018) and MINIGRID (Chevalier-Boisvert et al., 2023). Given a set of options and their execution data from the first room of MONTEZUMASREVENGE, we show that our algorithm can successfully transfer those options to other, visually dissimilar rooms. In MINIGRID, we demonstrate how our agent learns transferable options and uses them to rapidly solve a challenging exploration problem.

## 2 BACKGROUND AND RELATED WORK

We consider Markov Decision Processes (MDP) $M = (S, A, r, p, \gamma)$ where $S$ denotes the state-space, $A$ is the action-space, $p(s_{t+1}|s_t, a_t)$ denotes the transition dynamics and $r(s_t, a_t)$ is the scalar reward function. The RL objective is to find a policy $\pi(a|s)$ that maximizes the cumulative discounted reward over the lifetime of the agent.

An **option** $o$ is a temporally extended action and, for this work, is defined by a tuple $(I_o, \pi_o, \beta_o)$. The initiation set, $I_o : S \rightarrow \{0, 1\}$, is the set of states in which option $o$ can initiate. The termination set, $\beta_o : S \rightarrow \{0, 1\}$ is the set of states in which option $o$ successfully terminates. The option policy $\pi_o : S \rightarrow A$ is a controller that transitions the agent from states in $I_o$ to states in $\beta_o$.

**Identifying Sub-goals.** For this definition, the option termination set is analogous to a sub-goal. Identifying sub-goals is an important aspect of hierarchical reinforcement learning. One popular method by Andrychowicz et al. (2017) is a simple but powerful technique of generating goals from previously visited states and has lead to several branch-off works (Pitis et al., 2020; Fang et al., 2019). Nair et al. (2020) learn a latent dynamics model to generate sub-goals and Florensa et al. (2018) focus on selecting solvable sub-goals. Bagaria et al. (2023) generate a protogoal space which consists of many possibly useful sub-goals and an adaptive function to map this space to a small set of useful goals. All these methods focus on the initial discovery of sub-goals while our work focuses on transferring an existing sub-goal and so can be used in tandem with any of these methods.

**Learning transferable skills.** A task is defined by a goal an agent must achieve in the environment. For example, a task can be navigating to a location or obtaining an object. Sub-goals can exist along the trajectory towards the goal, which are intermediate goals that must be first achieved before a final goal can be completed, or a set of unrelated goals the agent is tasked with completing. An option can be used to achieve these sub-goals, by setting the termination set to be a sub-goal. The use of options promises to aid in task learning as we only need to learn to reach each sub-goal once but may need to reach this sub-goal many times in future. However, in their most basic formulation, options do not generalize. This is because an option is defined over a specific subset of the state space and has no prior knowledge about what features are needed for successful transfer. One approach to transfer is via derived input spaces with transferable semantics. Konidaris & Barto (2007) showed that an agent-centric representation, analogous to an egocentric space, would be sufficient for option generalization but requires a hand-designed agent-centric input. Dayan (1993) and Barreto et al. (2018) use successor features to learn policies that generalize across reward functions but not changes in transition dynamics. Touati et al. (2022) expand on the successor features framework to learn rapidly generalizing policies under changing reward dynamics. Other attempts at skill transfer learn from demonstration (Konidaris et al., 2011; Ranchod et al., 2015), using demonstrations from several tasks to identify and learn common skills. Wang et al. (2014) require first learning a large number of skills and retroactively compressing these skills into a single transferable skill and sub-goal. Barreto et al. (2019) combine previously learned options into new options, called the option keyboard, which

allows for combining previously learned, possibly irrelevant, skills into something more beneficial. A common approach to skill generalization Agarwal et al. (2021); Frans et al. (2017); Gomez et al. (2022) assumes access to a distribution of relevant tasks that can be sampled from during training. These methods leverage this distribution to either learn useful state representations or general policies and sub-goals, however, carry an implicit assumption that an expert has predefined which tasks should be generalized to. Many of these works require either a large library of skills or assume a distribution of relevant tasks. As such, these methods either delay the gain of generalized skills or assume the skill has already been predefined and so are not compatible with existing skill discovery methods. These methods are retroactive, looking backwards on what has been learned or provided. We wish to generalize *forward*—transferring a sub-goal from a single option instance.

**Unsupervised Representation Learning.** Representation learning is often used to compress high dimensional data to aid future learning (Bengio et al., 2013). One method to learn a useful representation is leveraging auxiliary tasks such as Denton et al. (2017) who use the downstream tasks such as classification and video prediction to learn a compact representation. Higgins et al. (2017) learn a disentangled representation of the state space to improve an agents domain adaptation. Representation learning has also been used to represent goals to improve exploration (Laversanne-Finot et al., 2018) and sample efficiency (Nair et al., 2018). Our work differs from these by learning multiple hypotheses, each of which is a unique state representation.

The **Diversity-By-disAgreement Training (D-BAT)** algorithm from Pagliardini et al. (2022) is designed to learn a classifier able to transfer to out-of-distribution data. This is done by making use of both labeled and unlabeled data in a semi-supervised setting using the loss function defined in Equation 1.

$$L = \sum_{m=1}^{N} (L_{label}(h_m) - \alpha \sum_{i=1}^{N} (\log(p_{h_i}^y \dot{p}_{h_m}^{\not{y}} + p_{h_i}^{\not{y}} \dot{p}_{h_m}^y))) \tag{1}$$

Where $L_{label}$ is the chosen label loss, $h_i$ is an ensemble member, $\alpha$ is the diversity coefficient and $p_{h_i}^y$ is the probability of class $y$ or class $\not{y}$. The labeled data is sampled from a subset of the distribution while the unlabeled data is collected from the full distribution. D-BAT is an ensemble method, which encourages each ensemble member to lower classification loss on the labeled data while encouraging disagreement among members on the unlabeled data. The members of the resulting ensemble each maintain a unique hypothesis over the relevant features. Given a sub-goal, we generate a set of labeled data with the intention of generalizing this option to unseen contexts.

## 3 LEARNING TRANSFERABLE SUB-GOALS

Hierarchical reinforcement learning simplifies complex problems by splitting a single task into multiple sub-tasks, each of which can be defined by a sub-goal which can be provided to the agent or discovered through a skill discovery algorithm. These sub-goals are used to learn a policy to complete the corresponding sub-task, giving rise to a skill. These sub-goals can be represented in many ways, for example a sub-goal to reach the door of a room can be defined over the agent's $(x, y)$ location or the agent's distance from the door. While both feature sets can be used to effectively learn an option in the discovering context, the agent's position sub-goal will not generalize to new states. Many previous works tackle skill transfer retroactively, learning a large collection of skills that are then compressed into a small collection of reusable skills or assuming a predefined distribution of relevant tasks. These skills are either found after countless interactions with the environment— delaying access to a transferable skill for a long time and still need to answer the question of when to group a collection of skills—or require an expert to define a set of tasks. Instead, we propose generalizing *forwards*, making these skills available to the agent after their initial discovery and allows us to leverage the rich collection of existing skill discovery algorithms.

Sub-goals are commonly modeled as classifiers. The classifier returns 1 for sub-goal states and 0 otherwise; when executing a skill the agent runs a policy until reaching a sub-goal state, where it receives a reward. This is the most important component of the skill and the output of most skill acquisition algorithms. With only a transferable skill policy, it is not clear when this policy has completed. A transferable initiation set only tells us some policy can be started to reach some sub-goal but this information alone has no value. However, if the agent has access to a transferable

termination set, the agent can learn a policy and initiation set that achieves the same sub-goal in a new context.

### 3.1 HYPOTHESIS GENERATION

Given state space $\mathcal{S}$, we sample a dataset $\mathcal{D}$ which defines a space of functions $\mathcal{F}$, consisting of all functions that accurately fit all points in $\mathcal{D}$. This space of functions is very large as there are many ways to define a function that accurately fits some set of points. A function's ability to generalize is measured by how accurately the function can predict data points that have not been included in $\mathcal{D}$. When learning a classifier to approximate one of these functions, the burden of generalization is placed on the quality of data collection; which must fully encompass the tasks you wish to generalize to in $\mathcal{S}$. If the collected data does not accurately represent the full distribution of relevant contexts, the classifier cannot generalize reliably. Unfortunately, this is precisely the scenario RL agents find themselves in when generalizing forwards.

We therefore propose that, instead of sampling a single function from $\mathcal{F}$, we can instead sample multiple functions, which form a collection of *hypotheses*, $\mathcal{H}$. Each function, $h \in \mathcal{H}$, describes different features of $\mathcal{S}$ which accurately describe $\mathcal{D}$. By creating hypotheses over $\mathcal{D}$ we increase the chance of learning generalizable features. For example consider an agent learning to open a cupboard door and all training samples have been collected on the same cupboard which contains a red ball. One hypothesis attends to the red ball inside the cupboard—which is only visible if the door is open—while another learns to distinguish the location of the cupboard door when it is open or closed. Both of these hypotheses accurately describe the training data $\mathcal{D}$ but make use of different sets of features. To further improve the quality of $\mathcal{H}$, we can require that our sampling process focuses on diverse functions, ensuring each function $h$ differs meaningfully on the features of our data $\mathcal{D}$.

We leverage the D-BAT algorithm to ensure diversity among $h \in \mathcal{H}$, however, any algorithm that would result in a collection of diverse functions can be used instead. This algorithm makes use of labeled data $\mathcal{D}$ but also requires a set of unlabeled data $\mathcal{U} \in \mathcal{S}$ to train an ensemble of classifiers, taking the environment state as input and outputs 1 if the state is a sub-goal or 0 otherwise. Each ensemble member is encouraged to reduce labeled loss while also decreasing agreement on $\mathcal{U}$. As a result, we learn an ensemble which is our collection of hypotheses $\mathcal{H}$, with each ensemble member attending to different sets of features in $\mathcal{S}$, informed by $\mathcal{U}$. Given some discovered sub-goal, we collect labeled data which—due to the nature of discovered sub-goals—is collected from a small subset of states in the state-space. We can leverage the exploration required during policy learning to collect an unlabeled dataset which contains states in which the sub-goal is not defined.

### 3.2 OPTION LEARNING FOR MULTIPLE HYPOTHESES

Each hypothesis $h \in \mathcal{H}$ maps to a new sub-goal, each of which is transferable if applied to a relevant set of tasks. For example, consider Figure 1. This figure shows three states in an environment, the

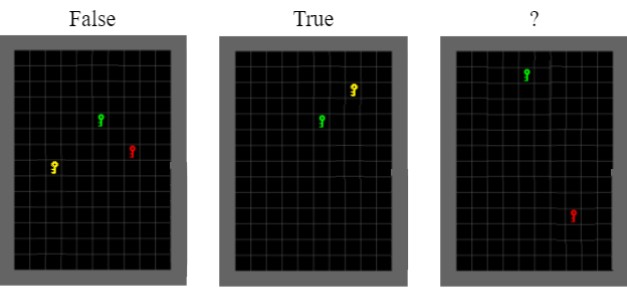

Figure 1: Given that the first image is *False* and the second image is *True* is the third image *True* or *False*?

first image is outside the defining sub-goal while the second is inside the sub-goal. Given these

two images, what is the sub-goal these images define? One possible hypothesis is that the red key must be removed, which accurately describes the given data. We can raise another hypothesis; the yellow key must be at a specific grid location. Another possible hypothesis may be that two keys are present at all times, regardless of color or location. All three of these hypotheses are consistent with the training data. Each of these hypotheses can be used to learn a distinct skill policy, one which collects the red key, one which places the yellow key at a specific grid location and one that ensures two keys are always present. As such, we learn a separate option policy for each ensemble member as all members correspond to valid sub-goals which may not overlap.

### 3.3 Hypothesis Selection

Once we have learned $\mathcal{H}$, we must select the best hypothesis. This task however is challenging and cannot be done without additional information. Returning to the example in Figure 1, can you determine if the third state satisfies the original sub-goal? The answer is it depends; the red key is present, so our red key sub-goal is not satisfied. Similarly the yellow key has been collected so our yellow key sub-goal is also not satisfied. There are, however, two keys in the environment so our two key sub-goal has been met. Which hypothesis is correct? If we were an agent in the environment, how do we decide which hypothesis is most beneficial? It is very likely that only one of these hypotheses is transferable given our task, however, without knowledge of what that task is, we cannot confidently select the most general hypothesis. But in an RL problem, the task is encoded in the environment reward. Therefore, we propose that the most transferable hypothesis will lead to higher external reward. We provide a high-level policy with an action space that contains options defined by all $h \in \mathcal{H}$, providing one option for each hypothesis. This high-level policy then learns to maximize the extrinsic reward, which encodes the task. As this high-level policy begins reward maximization, it will naturally begin selecting hypotheses that best support transfer.

## 4 Results

Our experiments are designed to first individually validate each component of our algorithm, and finally to test a consolidated RL agent on a challenging sparse-reward problem. Specifically, we seek to answer the following questions through our experiments:

1. Does the D-BAT algorithm produce accurate classifiers? Do the different ensemble members attend to different, diverse parts of the input-space?

2. How much *labeled* training data is needed to successfully generalize an option's sub-goal classifier to previously unseen contexts?

3. Are sub-goals generated by the D-BAT ensemble sufficient for learning useful option policies?

4. Can an agent equipped with transferable options learned using our algorithm solve challenging reinforcement learning problems?

5. Does the reward-maximizing high-level policy eventually prefer transferable options over non-transferable ones?

We define all sub-goals using expert collected trajectories to ensure we can accurately evaluate the quality of transfer during all experiments.

### 4.1 D-BAT Evaluation for Sub-Goal Generalization

We begin by confirming that each ensemble member is learning a unique hypothesis for a single sub-goal. MONTEZUMASREVENGE is an Atari game in which the player must traverse through several rooms, collecting treasure or keys to open future doors while avoiding enemies and hazards. Each screen in the game has a unique layout and looks visually different, making sub-goal generalization challenging from pixel input. All classifiers are given a gray-scale stack of the four previous frames resized to size $84 \times 84$. We begin by providing the D-BAT ensemble with labeled data from the starting room, which contains three ladders, and unlabeled data from one other room containing a ladder and one room without a ladder. We then use DeepLift (Shrikumar et al., 2017) to identify the most important features for each ensemble head for an unseen state. From Figure 2 we see these

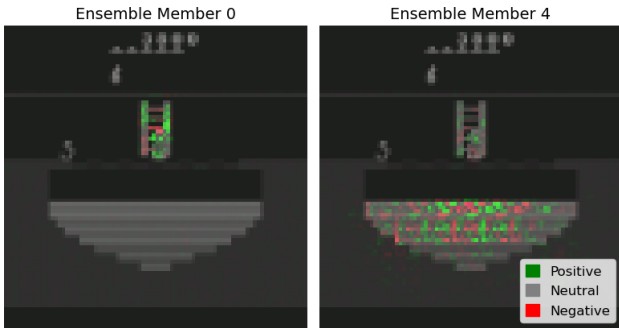

Figure 2: Saliency map showing that different ensemble members attend to different parts of the same image. Pixels marked as positive contribute to the classifier's final prediction; negative features contribute to the class the classifier did not predict. For simplicity, only one frame is shown. The attribution plot for all frames for each ensemble member can be seen in Appendix A Figure 9.

ensemble members have attended to different features of the image. Ensemble member 0 focuses on the ladder and player while ensemble member 4 has mainly attends to the lava below the floor. In this example, attending to the lava would not prevent the classifier from correctly identifying this sub-goal but will prevent generalization to future ladder instances if no lava is present.

Next we evaluate the accuracy of the classifiers produced by the D-BAT ensemble as well as how much labeled data is required for successful sub-goal generalization. MONTEZUMASREVENGE has multiple ladders located throughout the game, each in a visually distinct room. We define the CLIMBDOWNLADDER sub-goal as met when the agent is at the base of any ladder. The purpose of learning an ensemble of classifiers is to obtain at least one classifier defined over a generalizing set of features. Therefore, only the accuracy of the best performing ensemble member need to be considered. We compare against a one-headed ensemble, or standard CNN because traditionally sub-goals are represented with a single classifier providing a clear comparison against what is often done in practice and is equivalent to sampling a single hypothesis. We also evaluate the D-BAT ensemble with a diversity weight of 0 to show the effect of diversity on hypothesis sampling. Note that both ensembles represent a collection of hypotheses, however, one is trained to explicitly encourage diversity while a standard ensemble must rely on random initializations for diversity.

Figure 3 shows the accuracy of all classifiers improves greatly after seeing two rooms. However, because we intend to generalize forwards, the performance of the ensemble when provided labeled data for a *single ladder instance* is most significant as a skill discovery algorithm will only provide one example. For this case both ensembles outperform the CNN by around $20\%$. While the diverse ensemble has lower variance, we find both ensembles seem to perform equally well when evaluating on a set of previously collected data. We should note that because the state comprises of four previous frames, our current state is dependent on the previous four actions, making the space is very large. As such, our collected data does not fully encompass all possible states which an agent may encounter during policy learning, therefore, a high classification accuracy may not result in a useful option. For example, if a $90\%$ accuracy classifier includes the top of the ladder as part of the sub-goal, the resulting policy will transport the agent one or two steps down the ladder which is not a useful CLIMBDOWNLADDER skill.

### 4.2 D-BAT EVALUATION FOR POLICY LEARNING

It was previously shown that the D-BAT algorithm results in an accurate and diverse ensemble but this may not be enough for effectively training future skill policies. We again use the CLIMB-DOWNLADDER sub-goal, incrementally evaluating on larger labeled datasets, while now including skill policy learning. The agent is placed at the top of each ladder instance in the game, training a skill for each ensemble member for $3 \times 10^5$ environment steps per ladder. Each policy is given a reward of 1 if the sub-goal is satisfied and 0 otherwise. A new policy is trained for each ladder and the state is equivalent to the classifier input. After policy training, we average the Manhattan distance between the final termination location and the closest true termination for the last 100 option runs

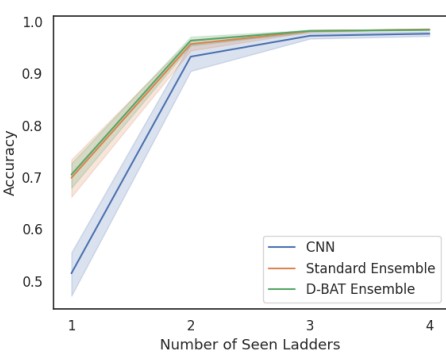 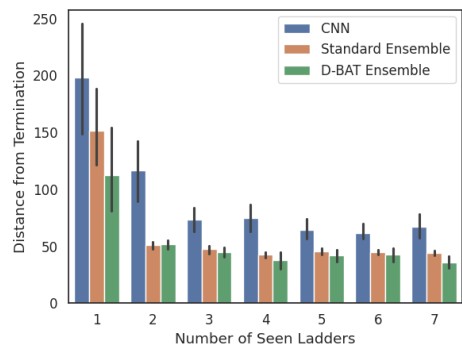

Figure 3: Accuracy of the best performing ensemble member as the ensemble is given labeled data from more ladders across the Montezuma's Revenge game. Results are averaged over 10 seeds and bands represent the standard deviation.

Figure 4: Average Manhattan distance between the state in which option execution terminated and the sub-goal region of the option; bars represent mean and standard deviation over the last 100 option executions (lower is better) averaged over 10 seeds.

and again average the performance for each ladder to get a single distance. Because we use a frame skip of 4, the Manhattan distance is equivalent to 4 times the number of actions away from the true termination. If an option starts in the termination set—the classifier predicts the top of the ladder as part of the sub-goal—the Manhattan distance is set to 300 which is the mode ladder length. We again consider only the best performing head for evaluation.

Figure 4 shows that, although the ensemble accuracies are very close, there is a significant difference in the Manhattan distance for the 1 ladder case. We see that the ensembles perform similarly when more data is available, indicating that while a diverse ensemble helps when we have limited data, there is no additional benefit once we have a diverse set of training data. However, we must keep in mind that we wish to generalize forward from one option instance and so cannot guarantee diverse data. For all instances we find the ensemble methods outperform the single CNN by a significant margin indicating that having multiple hypotheses improves sub-goal transfer.

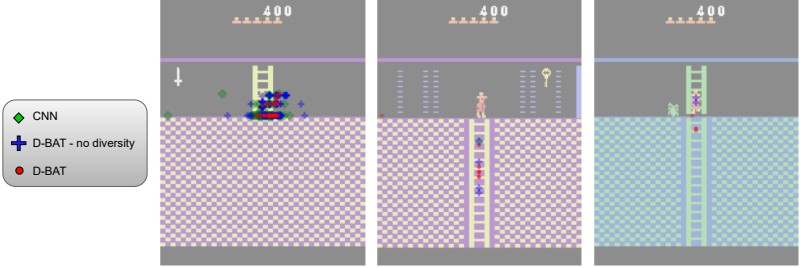

Figure 5: Scatter plot of termination locations. These are the locations of the agent for the last 100 skill executions for a single sub-goal and policy seeing data from two ladders. The agent starts on top of the ladder in the middle room. If the agent climbs down the ladder, they will arrive in the room to the right while traveling left from the starting room will place the agent in the left room. Only the best performing member is shown. We see the ensemble methods outperform the CNN, with the D-BAT ensemble resulting in more compact termination sub-goals.

While the Manhattan distance decreases as more ladder instances are introduced, it does not intuitively confirmed that our learned CLIMBDOWNLADDER policies are useful. To further investigate the quality of the sub-goals defined by the D-BAT ensemble, we plot the final 100 termination locations for the best performing member after exposure to two ladders on a previously unseen partition of the environment. Looking at Figure 5, we see both ensemble methods terminate at the base of

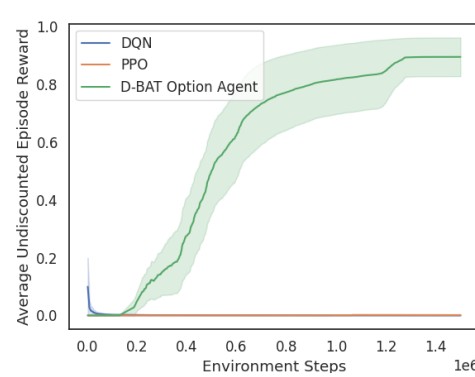

Figure 6: Average undiscounted reward for the modified MINIGRID DOORKEY environment. All results are averaged over 10 seeds and bands represent the standard error.

the leftmost ladder, with the diversity ensemble leading to a more compact termination set. The ensemble methods also identify the rightmost ladder, which continues through the floor of the room. During training, the classifiers see no labeled instances of ladders continuing through the floor as we leave these states unlabeled, showing that both ensembles have generalized to this case without seeing similar labeled examples. The CNN never identifies the right ladder termination and is less successful at identifying ladder bases despite having an accuracy of $90\%$ on our expert collected data. Both ensemble members also rarely terminate erroneously in the middle of the ladder. From these experiments we can conclude that both ensemble sub-goals can be used to learn policies that transfer the initial option, however, the D-BAT ensemble has significantly better performance.

### 4.3 CONSOLIDATED REINFORCEMENT LEARNING AGENT IN MINIGRID

Although, we previously showed that a collection of hypotheses can be used to learn option policies that transfer to new ladder instances in MONTEZUMASREVENGE, this is not significant unless these transferred sub-goals can be used by an agent to solve reinforcement learning problems. MINIGRID is a 2-dimensional grid-world environment consisting of goal-oriented tasks. The DOORKEY environment places the agent in one of two rooms separated by a locked door. The agent must collect the key to unlock the door and travel to the goal location in the other room. Traditionally, this environment has a red door and a single red key. We modify this task to include a blue and green key to increase the difficulty of the task. The agent must collect the correct key—the red key—to unlock the door and reach the goal to receive a reward of $1$ or $0$ for every other time step, making this a sparse reward problem. We use the full RGB image as our state.

We define 5 sub-goals; COLLECTBLUEKEY, COLLECTGREENKEY, COLLECTREDKEY, OPENREDDOOR and GOTOGOAL. Labeled data is collected from seed 0 and unlabeled data is collected from seeds 1 and 2. Data from test seeds is not shared among agents. The lower-level ensemble policies are not pre-trained, and must be learned in conjunction with the higher-level policy. We use a PPO (Schulman et al., 2017) agent with access to only the option policies, each of which is a Deep-Q Network (Van Hasselt et al., 2016) with a prioritized replay buffer (Schaul et al., 2015). Each ensemble member is an executable skill, with an independent low-level policy. The PPO agent has an action space of size 15, consisting of 3 hypotheses for all 5 original sub-goals. We aim to show that giving the agent skills that are not aligned with the overall task will not irreparably damage policy learning, and the strong performance of our ensemble agent cannot be explained by our use of DQN and PPO. As such we evaluate against a DQN and a PPO agent, each of which have access to the original environment actions. All agents are run for $1.5$ million environment steps with a maximum episode length of $500$ steps.

Figure 6 shows the option agent is able to learn to reach the goal consistently, while the DQN and PPO agents fail to complete the task. We can conclude that hypothesizing helps the agent complete the task goal by allowing the agent to leverage previously defined useful sub-goals, simplifying the original task.

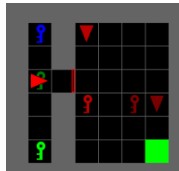 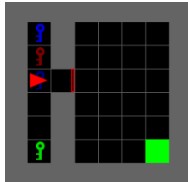

(a) Ensemble member 2 which is the least picked by the agent.

(b) Ensemble member 3 which is the most picked by the agent.

Figure 7: Overlaid termination states identified by members of the D-BAT ensemble for the OPENREDDOOR sub-goal.

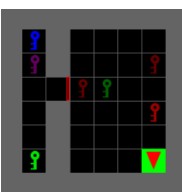 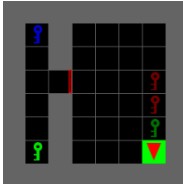

(a) Ensemble member 1, tied for most picked.

(b) Ensemble member 3, tied for most picked.

Figure 8: Overlaid termination states identified by members of the D-BAT ensemble for the GOTO-GOAL sub-goal.

### 4.4 HYPOTHESIS SELECTION THROUGH REWARD OPTIMIZATION

Our previous experiments have shown that we can learn *at least one* generalizing hypothesis but has not yet shown how to identify the corresponding ensemble member. We previously claimed that reward maximization is a useful proxy for identifying the hypothesis that best transfers. To evaluate this claim, we compare the termination sets of the most and least selected ensemble members. Overall, we find that the most popular hypothesis closely resembles our hand-defined generalized sub-goals. Figure 7 overlays multiple termination sets for two members of the OPENREDDOOR sub-goal. For the least selected ensemble member, the sub-goal does not fully correlate with our definition of the OPENREDDOOR sub-goal; i.e. positioned in front of an open door. On the other hand, Figure 7a shows the most picked ensemble member which requires the agent always stands at the open door, matching our hand-defined OPENREDDOOR sub-goal.

For the GOTOGOAL sub-goal, shown in Figure 8, we show two ensemble members, neither of which seems to be favored. Both of these sub-goals require the agent to be standing on the goal position which again aligns with our previously defined sub-goal. The final member of this ensemble, which has a significantly lower selection probability, does not trigger for this DOORKEY environment seed. This supports our statement that reward maximization leads to the selection of the most generalized sub-goals. We can conclude that, not only does diversity allow for successful sub-goal generalization for a given option but also that reward maximization is a good measure for identifying the most transferable hypothesis.

## 5 CONCLUSION

In this work we tackle sub-goal generalization from a single instance; the most crucial step for forward skill generalization. We do this by learning an ensemble of diverse hypotheses over generalizing feature sets using the D-BAT algorithm. Our experiments show that this ensemble can successfully transfer the CLIMBDOWNLADDER sub-goal in MONTEZUMASREVENGE with 70% accuracy when exposed to a single set of ladders in the game. These classifiers can be used to learn options that takes the agent to the bottom of ladders throughout MONTEZUMASREVENGE. We also show that these sub-goal classifiers can be used for learning a modified DOORKEY task in MINIGRID and reward maximization can be used to identify the hypothesis that best supports transfer.

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

## A ADDITIONAL RESULTS

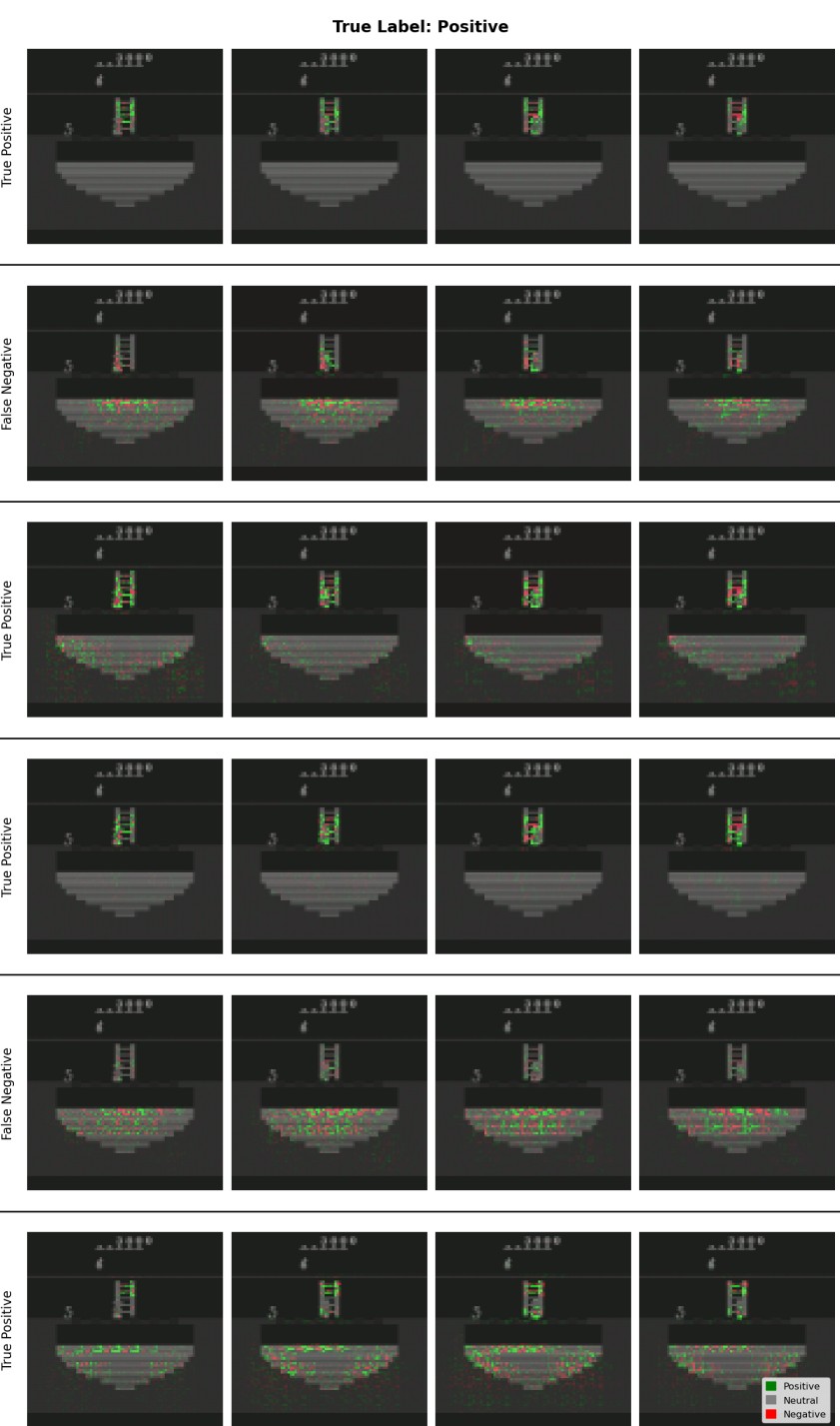

Figure 9: Feature importance of a sample for all ensemble members. Features marked as positive contribute to the classifier's final prediction. Negative features contribute to the class the classifier did not predict. We also include the label and predictions for each ensemble member.

## B  HYPERPARAMETERS

### B.1  MONTEZUMA'SREVENGE HYPERPARAMETERS.

All classifier models consist of three CNN layers, with batch normalization and ReLU activations between. The CNN layers then feed into two dense layers before final classification. We use the Adam optimizer and a cross entropy classification loss for labelled data. Unlabelled data is optimized using the Kullback-Liebler divergence.

Table 1: MONTEZUMA'SREVENGE D-BAT Ensemble Hyperparameters.

| Hyperparameter | D-BAT | D-BAT - no diversity | CNN |
|---|---|---|---|
| Learning Rate | $5 \times 10^{-4}$ | $5 \times 10^{-4}$ | $5 \times 10^{-4}$ |
| Diversity Weight | $3 \times 10^{-4}$ | 0.0 | $3 \times 10^{-4}$ |
| Member Number | 6 | 6 | 1 |
| L2 Regularization Weight | $5 \times 10^{-4}$ | $5 \times 10^{-4}$ | $5 \times 10^{-4}$ |
| Class weighting | 0.8 negative 0.2 positive | 0.8 negative 0.2 positive | 0.8 negative 0.2 positive |
| Batchsize | 64 | | |

We use a Deep-Q network for all policy training. The network consists of two CNN layers, each followed by a batch normalization layer, ReLU activation and max pooling layer. This is fed to a Gated Recurrent Unit before passing through two dense layers for action selection, optimized using the Adam optimizer. All options were allowed to run for 200 time steps before terminating if they did not reach the sub-goal defined by the classifier.

Table 2: MONTEZUMA'SREVENGE Policy Hyperparameters.

| Hyperparameter | Value |
|---|---|
| Replay buffer length | $1 \times 10^5$ |
| Update interval | 4 |
| Q-target update interval | 10 |
| Final Exploration frame | $4 \times 10^5$ decaying from 1 to 0.01 |
| Learning rate | $2.5 \times 10^{-4}$ |
| Batchsize | 32 |

### B.2  MINIGRID DOORKEY HYPERPARAMETERS.

All classifier models consist of two CNN layers, with batch normalization and ReLU activations between. The CNN layers then feed into three dense layers before final classification. We use the Adam optimizer and a cross entropy classification loss for labelled data. Unlabelled data is optimized using the Kullback-Liebler divergence.

We use a Deep-Q network for all policy training. The network consists of two CNN layers, each followed by a batch normalization layer, ReLU activation and max pooling layer. This is fed to a Gated Recurrent Unit before passing through two dense layers for action selection, optimized using the Adam optimizer. All options were allowed to run for 10 time steps before terminating if they did not reach the sub-goal defined by the classifier.

We train a PPO agent to reach the goal for DOORKEY MINIGRID. The PPO agent has three CNN layers reperated by ReLU activations fed into a dense layer. These layers are shared between the actor and critic layers. The actor has an additional gaussian policy head and the critic has a single dense layer.

Table 3: MINIGRID DOORKEY D-BAT Ensemble Hyperparameters.

| Hyperparameter | Values |
|---|---|
| Learning Rate | $2 \times 10^{-4}$ |
| Diversity Weight | $1 \times 10^{-4}$ |
| Member Number | 3 |
| L2 Regularization Weight | $1 \times 10^{-4}$ |
| Class weighting | 0.5 negative 0.5 positive |
| Batchsize | 64 |

Table 4: MINIGRID DOORKEY DQN Hyperparameters.

| Hyperparameter | Value |
|---|---|
| Replay buffer length | $1 \times 10^{5}$ |
| Update interval | 4 |
| Q-target update interval | 10 |
| Final Exploration frame | $8 \times 10^{3}$ decaying from 1 to 0.01 |
| Learning rate | $2.5 \times 10^{-4}$ |
| Batchsize | 32 |

Table 5: MINIGRID DOORKEY PPO Hyperparameters.

| Hyperparameter | Value |
|---|---|
| Replay buffer length | $1 \times 10^{5}$ |
| Update interval | 100 |
| Entropy coefficient | 0.01 |
| Lambda | 0.97 |
| Batchsize | 64 |
| Epochs per update | 10 |

