# OpenReview forum: "Learning Transferable Sub-goals by Hypothesizing Generalizing Features"
_ICLR.cc/2025/Conference — Submitted to ICLR 2025_

### Official Review · Reviewer_vzLt · 2024-10-29

**Soundness:** 3
**Presentation:** 4
**Contribution:** 2
**Rating:** 6
**Confidence:** 5

**Summary:**

This paper studies transferable skills and sub-goal generalization in the hierarchical reinforcement learning framework; the authors propose that the RL agent maintain several, diverse hypotheses over which features of the state might generalize in the future, focusing their attention on ensuring sub-goals are represented in a transferable way. The agent then selects from among these hypotheses, tests one of them in the environment, and updates its beliefs over which member of the ensemble has learned transferable features. Their work focuses on generalizing forward from a single option instance instead of retroactively compressing previously learned skills. Their work differs from previous related research by learning multiple hypotheses, each of which is a unique state representation. They apply these reusable sub-goals and perform an empirical study to MINIGRID and MONTEZUMA’S REVENGE environments, allowing agents to relearn previously defined skills in unseen parts of the state-space. Lastly, the authors seek several essential questions and tend to look for answers with their experiments and to test a consolidated RL agent on a challenging sparse-reward problem.
.

This paper is a magnificent algorithmic contribution to the hierarchical RL and reinforcement learning community in general and I strongly accept it as it is considered to be an authentic contribution to the ICLR conference.

The main contributions are the following:
- Forming each ensemble member as a unique hypothesis about the transferable features of a sub-goal that the agent can use to learn a skill in previously unseen portions of the environment.
- In contrast to previous methods that focus on the initial discovery of sub-goals their work focuses on transferring an existing subgoal and so can be used in tandem with any of these methods.
- Presenting sub-goals in a transferable way and empirically performing sub-goal generalization.

**Strengths:**

This paper does a great job of presenting novel authentic contributions. For originality, the paper tackles an important problem: How can learned features be transferable in HRL? The authors propose a very revolutionary concept;“hypothesis” to learn about the transferable features of a sub-goal that the agent can then use to learn a skill in previously unseen portions of the environment. Furthermore, their work builds upon and intersects with different methods: 1. Identifying sub-goals by focusing on transferring an existing subgoal and so can be used in tandem with any of the existing methods. 2. Learning Transferable Skills by generalizing forward from a single option instance instead of retroactively compressing previously learned skills. 3. Unsupervised Representation Learning by learning multiple
hypotheses, each of which is a unique state representation. The use of the Diversity-By-Disagreement Training (D-BAT) Algorithm by generating a set of labeled data representative of the option sub-goal with the intention of generalizing this option to unseen contexts. Eventually, the authors build brilliantly upon the RL hypothesis, i.e. the most transferable hypothesis will lead to higher external reward, providing a high-level policy. When this high-level policy maximizes extrinsic reward, it will naturally begin selecting hypotheses that best support transfer.


The significance of their work is integrating D-BAT which leverages both labeled and unlabelled data to train an ensemble of classifiers, each attending to a different set of features. Each ensemble member is encouraged to reduce the labeled loss while also decreasing agreement on the unlabeled data. As a result, each ensemble classifier represents a unique hypothesis of what features will generalize to out-of-distribution data, informed by the unlabelled data that was provided.

The empirical experiments were conducted to investigate and answer each question. The quality of their investigations is good and reflects each question considering both accuracy and the amount of labeled data required for successful sub-goal generalization as clearly shown in Figure 3. The authors succeed in validating each component of their algorithm. Additionally, experiments were done in MiniGrid DoorKey and MONTEZUMASREVENGE. Environments conclude that the D-BAT ensemble sub-goals can be used to learn policies that transfer the initial option as shown in Figure 5 and Figure 6. In general, this paper does a great job theoretically and empirically.

**Weaknesses:**

There seems no math or theorem that further explains your method or explains your formulation of "hypothesis" as a mathematical concept. This is particularly called into question due to the lack of math notations. In addition, no pseudocode was provided nor any clear statements about the connection with The Diversity-By-disAgreement Training (D-BAT) algorithm. More discussion on these areas would be much appreciated.

For the empirical study, the choice of baselines in the MONTEZUMA’S REVENGE Environment
was restricted to the CNN classifiers without any clear explanation which makes it unclear for the reader. Similarly, there was no discussion of the choice of baselines in the MINIGRID DOORKEY Environment.

Minor comments:
P.6, Line 1 “need to be” instead of “need be”.

**Questions:**

I am very interested in your formulation of “hypothesis” but at the same time was disappointed for not seeing enough explanation or theorem that covers your idea. How did you exactly formulate “hypothesis” in your code? Can you provide any theorem and/or math notations that support your formulation?

In what respect does your use of representation learning differ from using dimension reduction? How do you exactly provide the high-level policy?

In Figure 3, you mentioned that “this performance is on a collected set of data and does not fully encompass all the possible states an agent may encounter during policy learning.” How would the result differ from not using a collected set of data or any suggestions to overcome this? Why did you use the PPO and DQN agents to compare with? Similarly, why did you choose the CNN classifiers to compare with? A bit more discussion on these choices would be helpful.

---

> ### Author Response · Authors · 2024-11-15
>
> Thank you for your review. We are glad to see that you deeply understood our paper. We plan to revise the paper with a mathematical definition of a hypothesis as well as clarify some of our experiment decisions.
>
> > I am very interested in your formulation of “hypothesis” but at the same time was disappointed for not seeing enough explanation or theorem that covers your idea. How did you exactly formulate “hypothesis” in your code? Can you provide any theorem and/or math notations that support your formulation?
>
> We plan to update the paper with a formal definition. Given a dataset *D* we can define a space of functions *f* that accurately fit the data *D*. Our hypotheses are sampled from this space which is very large as there are many ways to fit a finite collection of points. Because a neural network is approximating a function, we represent each hypothesis as an individual neural network.
>
> >  In addition, no pseudocode was provided nor any clear statements about the connection with The Diversity-By-disAgreement Training (D-BAT) algorithm. More discussion on these areas would be much appreciated.
>
> D-BAT uses unlabeled data to sample a set of diverse functions from our hypothesis space. This is done by requiring that each function disagree on a set of unlabeled data. This disagreement ensures each hypothesis is varied in a significant way while maintaining consistency with the labelled dataset which is why we chose to leverage D-BAT.
>
> > In what respect does your use of representation learning differ from using dimension reduction?
>
> When performing dimensionality reduction you are always discarding information. A good reduction would remove irrelevant information while a bad reduction would also discard pertinent information. If you have a single subgoal to generalize from, it is not clear what features should be removed and which should be kept. By learning a diverse ensemble, each ensemble member discards different parts of information and so is performing a kind of dimensionality reduction.
>
> > How do you exactly provide the high-level policy?
>
> Our high-level policy is a PPO algorithm that receives the full state and must select an action, each of which is a low-level policy that is tasked with satisfying a single subgoal from the ensemble of classifiers. In the MiniGrid experiment, we started with 5 options and used a 3 head ensemble for each. The high-level policy had an action space of 15 actions, one action for each ensemble member for all 5 ensembles.
>
> > In Figure 3, you mentioned that “this performance is on a collected set of data and does not fully encompass all the possible states an agent may encounter during policy learning.” How would the result differ from not using a collected set of data or any suggestions to overcome this?
>
> In the Montezuma’sRevenge experiment we use a state composed of a stack of the previous 4 frames. As a result our state is now a factor of the previous 4 actions, atari has 18 possible actions, so there are a large number of possible states. We used expert collected trajectories for our data and so did not cover all possible states for every ClimbDownLadder subgoal. If this data was coming from an existing option, it is likely that there would be larger variation in these states as the policy executes random actions during its initial training. However, this cannot be guaranteed if we are allowing the agent to autonomously discover and train different skills and subgoals.
>
> > Why did you use the PPO and DQN agents to compare with? Similarly, why did you choose the CNN classifiers to compare with? A bit more discussion on these choices would be helpful.
>
> We agree that more discussion on these choices is needed and will be adding our motivations in the paper revision. Our ensemble agent uses DQN for the lower-level option policies and PPO for the higher-level policy. Our motivation for testing against these algorithms was to definitively show that the performance of the ensemble agent was not due to the power of these deep RL methods but instead our transferred subgoals.
>
> For the Montezuma’sRevenge experiments we choose to test against a CNN because this is traditionally how subgoals would be defined and so clearly shows how our method performs against current techniques as well as highlighting the difficulties for current CNN based methods.
>
>
> We would again like to thank the reviewer for their thoughtful review and will address their concerns in our revision. We are happy to address any further questions or concerns.

---

> > ### Author Response · Authors · 2024-11-17
> >
> > Dear reviewer,
> >
> > We have updated our paper draft with the following changes
> >  - Clarified our goal in the introduction
> >  - Added additional related works (mostly to the Learning generalized skills section)
> >  - Clarified the algorithm methodology, adding mathematical context for hypotheses and improving descriptions of how D-BAT fits into the algorithm
> >  - Clarified experimental set-up and decisions
> >
> > We will be updating our experiments with additional seeds once all runs have completed.
> >
> > Please let us know if you have any further questions or concerns.

---

> ### Author Response · Authors · 2024-11-21
>
> Dear reviewer,
>
> We have updated all experiments to be averaged over 10 seeds.
>
> We believe we have addressed all your concerns and are eager to engage with you on any further points or questions you may have!

---

### Official Review · Reviewer_2Yzk · 2024-10-31

**Soundness:** 2
**Presentation:** 2
**Contribution:** 2
**Rating:** 3
**Confidence:** 4

**Summary:**

The paper argues for the use of an existing technique called DBAT to learn ensembles of state binary classifiers in the context of Reinforcement Learning (RL). The use of each classifier is to encode a goal, understood as a set of states, which can be used to train an option policy. Relying on the motivation for DBAT, the paper explains and motivates empirically the use of an ensemble of classifiers, rather than a single one. In particular, they show that the ensemble classifiers do generalize to unseen states and that option policies can indeed be learned from them in the environment Montezuma's Revenge, while this is not the case for single classifiers (referred to as CNNs in the text). Moreover, they show how a hierarchical use of the options learned from   an ensemble of classifiers can solve a MiniGrid task where standard end-to-end deep RL algorithms fail.

**Strengths:**

Originality (and significance):
- The paper highlights an important and typically overlooked aspect of the problem of learning transferable state and goal representations in RL: it is not possible to tell what information is useful to transfer and for this reason multiple hypothesis, or a similar flexible representation, should be used, as opposed to a single standard architecture that tries to identify "the best" features.
- The paper introduces the use of a non-standard method to learn state classifiers in RL.

Quality:
- The paper provides multiple pieces of empirical evidence (three, to be exact) to demonstrate that the chosen technique, DBAT, is useful to encode goals, or, more generally, sets of states.

Clarity:
- The organization of the paper is good. The problem and its motivation are well motivated and the exposition of the proposed solution follows a logical order.
- The Results section contains an explicit list of questions to be answered and it does attempt to explicitly answer each and every one of them.

**Weaknesses:**

The Introduction states that "To fully realize the benefits of HRL, learned options should be transferable. (...) existing HRL methods are unable to generalize an option from one context to another. This is primarily because all three components of the option are conditioned on the entire state, which includes spurious features unnecessary for successful execution." From this, I identify a main goal and two claims. The goal is to provide a scalable method to learn transferable options, and the claims are that no HRL method does this, and that the reason for why they fail is that, whatever methods are being used to learn option policies, they result in architectures that rely on spurious correlations to components of the input sensory stream. In my opinion, the goal is not achieved and the claims are not supported.

Why is the main goal not achieved?
1. There is no mechanism to discover the goals. The images used to train DBAT were hand-picked. How can this process work in more general settings?
2. It is necessary to train preliminary options to generate rich enough data to train the ensemble classifiers. Where do these options come from? How can we guarantee that they will generate rich enough data? In this case they were clearly well chosen, e.g., the MiniGrid environment case contains the three option policies that are required to complete the general task (CollectRedKey,
OpenRedDoor, and GoToGoal). This begs the additional question. Having those options already, why bother to use a classifier to learn options that carry out the same task as the original options?
3. The way in which the option policies are obtained from the classifiers is unclear and possibly does not scale to more general settings. In particular, for the Montezuma's Revenge environment, the manuscript mentions that an agent is manually placed at the top of some stairs and then it is supposed to learn to go down the stairs. What happens when the agent is not placed on top? More generally, what happens in other tasks where there is no control over this initial state?
4. There is no evidence of transfer under different reward functions or dynamics. For MiniGrid, the option GoToGoal arguably already encodes a lot of information about the original reward function.

Why are the claims not supported?
1. The paper ignores relevant literature in reward-agnostic option discovery, representation learning for RL, and multi-task reinforcement learning that partially solve the main goal addressed in the paper. For example:\
   a. Agarwal R., et al., Contrastive Behavioral Similarity Embeddings for Generalization in Reinforcement Learning, ICLR, 2021.\
   b. Barreto A., et al., Combining Skills in Reinforcement Learning, NeurIPS, 2019.\
   c. Eysenbach B. et al., Diversity is All You Need: Learning Skills without a Reward Function, ICLR, 2019.\
   d. Frans K., et al. Meta Learning Shared Hierarchies, ICLR, 2018.\
   e. Gomez D., et al. Information Optimization and Transferable State Abstractions in Deep Reinforcement Learning, IEEE TPAMI, 2022.\
   f. Klissarov, M. et al., Deep Laplacian-Based Options for Temporally-Extended Exploration, ICML, 2023.\
   g. Touati, A. et al., Does Zero-Shot Reinforcement Learning Exist?, ICLR, 2023.\
   h. Zhang A., et al., Learning invariant representations for reinforcement learning without reconstruction, ICLR, 2021.\
More specifically, a., e., and h. provide techniques to learn transferable representations, similar to the proposed ensemble of classifiers; b., c., d., and f. introduce techniques to learn transferable skills, and g., provides a global policy that solves any given task provided access to its reward function.
2. The paper makes no comparison with any other hierarchical method. Is there any advantage to the proposed approach?
3. The paper makes no empirical attempt to prove that the reason why any of the papers mentioned in the Background and Related section fail is because of the poor state generalization.

**Questions:**

- How are the sub-goals being used to train the option policies? Do they define a sparse reward that is equal to 0 unless the goal set is reached? How can you guarantee that learning this is not just as difficult as solving the original task?
- For MiniGrid, did you try learning a high level policy with the options that were used to train the ensemble classifiers? If so, how does it compare to the one using the classifiers?
- For MiniGrid, in total how many interactions did it take to train the original 5 options and then the 15 option policies corresponding to the ensembles? What happens if you run DDQN or PPO for that many steps?
- For MiniGrid again, what are the standard techniques used to solve these tasks? If not DDQN or PPO, why did you select them? Sounds like this is a hard exploration problem, so RND of DCEO (Klissarov et al., 2023) could be used instead. If it is them, how did you select their hyperparameters?
- What is the false positive rate of the CNN? If it is small, how do you explain that the CNN has high accuracy and yet leads to bad option policies? To me, this means that there is some problem with your accuracy calculation that does not capture transferability.
- How exactly did you pick the selected data for the training of the classifiers? More explicitly, what type of procedure did you follow? It sounds cumbersome separating all the images generated from the interaction with the environment.
- The paper claims that the rightmost image in Figure 5 is evidence of the option generalizing the notion of going down the ladder, but it shows the opposite to me. It shows that the agent only learned to go down a few steps, a similar or smaller number of steps than when it goes down in the original shorter stairs. Am I interpreting erroneously the dots being displayed?

---

> ### Author Response · Authors · 2024-11-15
>
> Thank you for critical and insightful comments that expose weakness in our communication. It has been clearly stated that
>
> > The goal is to provide a scalable method to learn transferable options, and the claims are that no HRL method does this
>
> We agree that we have not achieved these goals. However, this was not the goal of our research, which we will make clear in our revision of the paper. Rather, our goal is to generalize subgoals from one previously given option instance. We agree that we do not perform goal discovery (c and f) because we intend to work with existing option discovery methods, which identify single, unique subgoals which are not reusable outside their discovering instance. This is a result of two problems a) the subgoal is defined by a collection of states, labeled as either inside or outside the subgoal which can only be sampled from the initial discovered subgoal leading to a very small dataset with very little diversity/coverage, and b) this dataset of samples does not provide the ability to discern features important for the subgoal as this is fundamentally underspecified (we argue this point with an illustrating example in section 3.2).
>
> The review correctly points out existing works that tackle skill generalization. These useful references will be added to our Learning transferable skills related works section. The works mentioned (a,b,d,e,g) as well as the works already referenced, approach skill generalization retroactively; these works either require large libraries of skills that can be compressed or changed to form generalized skills (b) or assume sampling from a distribution of tasks which implicitly assumes an expert has curated a collection that encompasses what the agent must learn (a,d,e,g). Our goal is instead to approach skill generalization in the *forward* direction, i.e. if we are given one example of the skill (which is what most subgoal discovery algorithms provide) how do we generalize this skill to future tasks which are likely in never-before seen states? If we wish to make use of the rich body of work on subgoal discovery, it is important to be able to generalize these skills from the single instance in which they are discovered by these algorithms.
>
> The representation learning work again differs from our scenario as it uses full episode trajectories to learn representations that allow for better learning by removing irrelevant features. However, from one example of a subgoal it is not possible to decide which features are truly irrelevant. Especially when considering the fact that what may be irrelevant to an agent (such as the color of an object during object manipulation) may not be irrelevant to a subgoal classifier. Thank you for bringing these works to our attention and will add them to our related works section.
>
> > There is no mechanism to discover the goals. The images used to train DBAT were hand-picked. How can this process work in more general settings?
>
> We hand-picked subgoals since our aim is to evaluate how well our method can generalize which requires knowing what our ensemble must generalize to. However the data used to train the D-BAT classifiers are what would typically be generated by an off-the-shelf skill discovery algorithm.
>
> > It is necessary to train preliminary options to generate rich enough data to train the ensemble classifiers. Where do these options come from?
>
> Since we intend to work in tandem with existing subgoal discovery methods, these preliminary options can come from any subgoal discovery method that generates subgoals.
>
> > How can we guarantee that they will generate rich enough data?
>
> The core problem is that we cannot guarantee this. In fact, we can almost always guarantee the opposite: since we are gaining data from a small collection of states, it is not rich enough to effectively train a traditional classifier to generalize. This is why we leverage multiple diverse classifiers which each attend to separate features from the provided data.
>
> > Having those options already, why bother to use a classifier to learn options that carry out the same task as the original options?
>
> These subgoals all represent options that we want the agent to continue to use. We define these subgoals only for seed=0, a single possible Minigrid DoorKey configuration. If we change the seed however, we want the agent to continue to use these options.
>
> > The way in which the option policies are obtained from the classifiers is unclear and possibly does not scale to more general settings
>
> We learn a policy for each ensemble member. The policy is given a reward of 1 if the relevant ensemble member returns 1 and 0 otherwise.

---

> > ### Author Response · Authors · 2024-11-15
> >
> > > What happens when the agent is not placed on top? More generally, what happens in other tasks where there is no control over this initial state?
> >
> > In the Montezuma’sRevenge experiment, an option execution is considered a success if any ladder bottom is reached. This means that the agent may not have climbed down a ladder to reach the base of a ladder, but could instead walk to an adjoining room that contains a ladder base. Additionally, in the case of the Minigrid DoorKey experiment the task had no control over the initial state. Each option could be executed at any point as long as the state did not already satisfy the subgoal. In these cases the policy learns to satisfy the subgoal, if possible.
> >
> > > There is no evidence of transfer under different reward functions or dynamics
> >
> > Our option policies are trained only to satisfy the corresponding subgoal classifier. As such we do not rely on the reward function. Each policy is trained from a random initialisation and so if we attempt to generalize to a subgoal in an environment with different dynamics our option policy will train as normal.
> >
> > > For MiniGrid, the option GoToGoal arguably already encodes a lot of information about the original reward function.
> >
> > This is true, however, given that the door is always red in our test environments, the CollectBlueKey and CollectGreenKey do not correlate to the environment reward.
> >
> > > The paper makes no comparison with any other hierarchical method. Is there any advantage to the proposed approach?
> >
> > We do not compare to subgoal discovery methods because our method is intended to work alongside these methods. We do not compare to existing skill generalization work as all the works we have found either require a library of prelearnt skills or a distribution of tasks from which to sample. Since our aim is to generalize from one subgoal these comparisons would be unfair.
> >
> > > The paper makes no empirical attempt to prove that the reason why any of the papers mentioned in the Background and Related section fail is because of the poor state generalization.
> >
> > We previously mentioned why existing skill generalization work differs from our research. Our classifier experiments compare against a CNN which is traditionally how a subgoal classifier would be represented in subgoal discovery methods. In Figure 3 and 4 we show a CNN struggles to generalize from a single option instance.
> >
> > > How are the sub-goals being used to train the option policies? Do they define a sparse reward that is equal to 0 unless the goal set is reached?
> >
> > We will clarify, in a revision to the paper, how these subgoals are used during policy training. Option policies are given a reward of 1 if the subgoal classifier returns true and 0 otherwise.
> >
> > > How can you guarantee that learning this is not just as difficult as solving the original task?
> >
> > There is no guarantee. Additionally it is possible that some ensemble members will learn subgoals that can never be met by the agent. In these cases the option does not correlate well to the overall task and so the hierarchical agent will stop selecting these options as they result in low environment reward.
> >
> > > For MiniGrid, did you try learning a high level policy with the options that were used to train the ensemble classifiers? If so, how does it compare to the one using the classifiers?
> >
> > We have trained an agent with perfect subgoal classifiers (note this agent has only 5 options as opposed to the 15 of the ensemble agent). This agent is able to learn the task faster than our ensemble agent, which is expected as there are fewer lower level policies to train.
> >
> > > For MiniGrid, in total how many interactions did it take to train the original 5 options and then the 15 option policies corresponding to the ensembles? What happens if you run DDQN or PPO for that many steps?
> >
> > The “initial 5 options” are described only as subgoals, no policies were trained in the original seed (seed=0). During testing the agent was not provided pretrained options and so was required to train the lower level policy in conjunction with the higher level policy which is the 1.5 million steps reported.

---

> > > ### Author Response · Authors · 2024-11-15
> > >
> > > > For MiniGrid again, what are the standard techniques used to solve these tasks? If not DDQN or PPO, why did you select them? Sounds like this is a hard exploration problem, so RND of DCEO (Klissarov et al., 2023) could be used instead. If it is them, how did you select their hyperparameters?
> > >
> > > We did not properly explain our selection of comparisons in the paper and plan to rectify this. Our ensemble agent uses DQN networks for the lower level policies and PPO for the higher level policy. We compare against pure DQN and PPO with the same hyperparameters because we want to make it clear that our ensemble agent is able to learn, not because of the power of these deep RL methods, but because of the use of our subgoals. Keep in mind that the ensemble agent is provided 15 options, of which only a handful may be useful, so we wished to determine if this large number of ill-fitting options would harm our agents.
> > >
> > > > how do you explain that the CNN has high accuracy and yet leads to bad option policies? To me, this means that there is some problem with your accuracy calculation that does not capture transferability.
> > >
> > > Our data is collected from a handful of expert trajectories. For Montezuma’sRevenge the input is a stack of the 4 previous frames. As such our state is dependent on our previous 4 actions, which considering atari has a total of 18 actions, may result in a large number of possible states for a single room and are not fully captured in our expert collected trajectories. During policy training, as the agents start by executing random actions, they are likely to encounter previously unseen states. This result shows that the CNN struggles to generalize to rooms it has seen if there are unique combinations of actions being executed which may not have been detected if testing is not done on every possible state.
> > >
> > > > How exactly did you pick the selected data for the training of the classifiers? More explicitly, what type of procedure did you follow?
> > >
> > > We used a handful (2-5) of expert trajectories designed to move the agent around the room but do not fully encompass the entire state space of the tasks.
> > >
> > > > The paper claims that the rightmost image in Figure 5 is evidence of the option generalizing the notion of going down the ladder, but it shows the opposite to me. It shows that the agent only learned to go down a few steps, a similar or smaller number of steps than when it goes down in the original shorter stairs. Am I interpreting erroneously the dots being displayed?
> > >
> > > First, we will better contextualize the images in the revised paper. The agent starts at the top of the ladder shown in the middle image. If the agent travels left it reaches the room depicted in the left image. If the agent travels down the ladder, it reaches the room to the right. It is true that the agent will erroneously terminate after a few steps down the ladder in some cases. However, all these points are taken from the same policy and subgoal classifier which is also able to travel past these erroneous states to the true termination depicted on the right. This same policy is also able to travel to the left most room to another true termination.These points are the position of the agent in the space of the game but do not fully describe the entire state of the agent, which is a function of the four previous frames, so it is possible for the same policy to travel the entire length of the ladder and erroneously terminate after a few steps depending on what state the agent is in. An interesting point to note is that the CNN is never able to get past the first ladder into the right-most room with the true termination.
> > >
> > > We will clarify the presentation to accurately convey the goal of our work and the setup of the experiments. We will inform you when these changes have been completed.
> > >
> > > Thank you again for your time and review and we look forward to any additional questions you may have.

---

> > > > ### Author Response · Authors · 2024-11-17
> > > >
> > > > Dear reviewer,
> > > >
> > > > We have updated our paper draft with the following changes
> > > >  - Clarified our goal in the introduction
> > > >  - Added additional related works (mostly to the Learning generalized skills section)
> > > >  - Clarified the algorithm methodology, adding mathematical context for hypotheses and improving descriptions of how D-BAT fits into the algorithm
> > > >  - Clarified experimental set-up and decisions
> > > >
> > > > We will be updating our experiments with additional seeds once all runs have completed.
> > > >
> > > > Please let us know if you have any further questions or concerns.

---

> ### Author Response · Authors · 2024-11-21
>
> Dear reviewer,
>
> We have updated all experiments to be averaged over 10 seeds.
>
> We believe we have addressed all your concerns and are eager to engage with you on any further points or questions you may have!

---

> ### Comment · Reviewer_2Yzk · 2024-11-24
>
> First of all, I want to thank the authors for replying to all the questions posed in my reply. They gave me some clarity.
>
> Second, I will explain my current decision regarding the score I assign to the paper and then provide some actionable suggestions.
>
> Decision:
>
> In my previous review, I identified a goal and two claims in the paper and I explained why in my opinion this goal is not satisfied and why the claims are not supported by compelling arguments or evidence. In response, the authors' reply that the identification of the goal is incorrect. They state that the goal is not to provide a method to learn transferable options, but instead a method that generalizes “subgoals from one previously given option instance”. However, I fail to clearly see the distinction between the two goals and, most importantly, I do not think the revised version makes any distinction between these different goals. In particular, the text says that “However, existing HRL generalization methods are unable to generalize an option from a single context to another.” This sentence seems to point out that the problem with previous techniques is that options learned are not transferable. This, to me, is exactly the goal that I previously identified and, given that neither the arguments nor the experiments changed, my score cannot change.
>
> That being said, I feel that I need to explain myself better. In particular, I will explain in what follows why I think the two goals are essentially the same, why the paper provides an interesting but neither novel nor complete solution, why it does not do any relevant comparison with similar techniques, and why it misses important details of previous work.

---

> > ### Comment · Reviewer_2Yzk · 2024-11-24
> >
> > Why are the goals the same?
> >
> > The paper assumes the pre-existence of a single option, which presumably is obtained via some standard RL technique like DQN or PPO. Then, it argues that, while the option might perform well in some initial task where it was learned, it does it by relying on partial information about the states or observations that, in general, will not be relevant for future tasks or states faced by the agent. The reason being that, while there might exist “generalizable features” (that is, an appropriate portion of the state information), they are unidentifiable. For this reason, the option policy itself should not be transferred, but its subgoal states should. The objective, then, is to find a way to extract and encode the subgoals of options in a manner that captures the generalizable features without having direct access to them. How are these subgoals supposed to be used posteriorly? The paper states that they should be used to learn many options, with the reasoning that at least one of them should be able to rely on the correct partial information. In this way, the implicit motivation that the paper gives to “generalize subgoals from one previously given option instance” is precisely solving the problem of finding transferable options, the goal that I initially identified.
> >
> > Why is the solution not novel?
> >
> > The solution proposed relies on the idea that knowledge is always incomplete and that this uncertainty should be considered in future tasks. To capture this idea, the paper introduces the concept of “forward” generalization. While I have not seen the same exact exposition and use of an ensemble to do this type of generalization, the idea is not new by any means in RL. In particular, works that use Bayesian, information theory, or robustness ideas like mixture of experts, ensembles, or the information bottleneck all use to some extent the same principle: that states encode transferable information that should be preserved while making the least amount of assumptions. See the following references for examples:\
> >     A) (Already mentioned in the review) Agarwal R., et al., Contrastive Behavioral Similarity Embeddings for Generalization in Reinforcement Learning, ICLR, 2021.\
> >     B) Castro P.S., et al., Mixture of Experts in a Mixture of RL settings, RLC, 2024.\
> >     C) Fan, J. et al., DRIBO: Robust Deep Reinforcement Learning via Multi-View Information Bottleneck, ICML, 2022.\
> >     D) Eriksson, H. et al, SENTINEL: Taming Uncertainty with Ensemble based Distributional Reinforcement Learning, PMLR, 2022.\
> >     E) (Already mentioned in the review) Gomez D., et al. Information Optimization and Transferable State Abstractions in Deep Reinforcement Learning, IEEE TPAMI, 2022.\
> >     F) Igl, M. et al., Generalization in Reinforcement Learning with Selective Noise Injection and Information Bottleneck, NeurIPS, 2019.\
> >     G) Mazoure B. et al., Deep Reinforcement and InfoMax Learning, NeurIPS, 2020.\
> >     H) (Already mentioned in the review) Zhang A., et al., Learning invariant representations for reinforcement learning without reconstruction, ICLR, 2021.\
> >     I) Lu, X. et al., Dynamics Generalization via Information Bottleneck in Deep Reinforcement Learning, arXiv:2008.00614, 2020.\
> >     J) Goyal, A. et al., InfoBot: Structured Exploration in Reinforcement Learning Using Information Bottleneck, ICLR, 2019.\
> > Islam, R. et al., Representation Learning in Deep RL via Discrete Information Bottleneck, AISTATS, 2023.
> >
> > Why is the proposed solution not complete?
> >
> > The authors and the paper imply that previous works are not acceptable since they assume access to pre-learned skills or a distribution of relevant tasks. However, in the shown experiments they make use of the same. In particular, in both Montezuma’s Revenge and MIniGrid, they assume the existence of an option, or multiple options, with a given goal set. In their case it comes from a human expert that labels images. The authors argue that in a more general setting the goal set would come from an off-the-shelf skill discovery algorithm. However, I find this answer unsatisfying. First of all, to my knowledge, most skill discovery algorithms find option policies, rather than full options (this is the case for most of the references I suggested that learn skills). Second of all, if we assume the existence of a subgoal discovery algorithm, what are the assumptions behind this algorithm working? If the algorithm is good enough to accurately classify states, why do we need an additional ensemble? If it is not, why should we expect the input data to the ensemble be sufficient to train it in a way that it encodes any useful information? The authors seem to believe that perfectly labeling the ladders, the keys, the door, or a final maze location has no implications on the performance of the algorithm, but I think that this is just as strong an assumption as it is assuming the existence of perfect options or selecting relevant tasks.

---

> > > ### Comment · Reviewer_2Yzk · 2024-11-24
> > >
> > > Why are there no relevant comparisons?
> > >
> > > The paper repeats in the first three sections that other algorithms are insufficient, yet at no point it provides any empirical evidence for this. Also, in their reply, the authors explain why the proposed algorithm should not be compared against any other algorithm. It should not be compared with goal discovery because it is supposed to work on top of goal discovery algorithms, and it should not be compared against other option discovery algorithms since it would be “unfair”. I disagree with both reasons. In the first place, it is not clear to me that the proposed ensembles are compatible with any goal discovery algorithm. This is not to say that the ensembles cannot work with goal discovery algorithms. What is not clear is if the ensembles provide any advantage over the existing goal discovery algorithms. As I mentioned earlier, if a good enough classifier of states exists to label them as goals or not goals, why is an ensemble necessary at all? The authors may point out that even a perfect classifier may not generalize to the new task, which I guess was the point of showing that accuracy does not perfectly correlate with future performance in Montezuma's experiments. To this, I would reply that training multiple options and learning a hierarchical algorithm over them might result in a lot of unnecessary interactions that might be better used for training online the goal discovery algorithm in the current task at the same time that a single option is trained with this perfect classifier tuned for the most recent task. Secondly, I am not convinced that algorithms that rely on the information bottleneck or similar concepts mentioned before do not implicitly solve the same problem of relying on spurious correlations without the need to train a large number of skills that will be ignored.
> > >
> > > Why does the paper miss important details of previous work?
> > >
> > > Most importantly, I did provide references of algorithms that do not rely on any initial set of pre-trained skills nor assume any expert selection of tasks but they were dismissed precisely because they do attempt to solve the problem of option discovery. Namely, references c) and f) are reward-agnostic, thus there is no selection of tasks, and they only need an initial random policy, hence no need for a large number of pre-trained skills.
> > >
> > > Furthermore, the revised version of the paper states that “Barreto et al. (2019) require first learning a large number of skills and retroactively compressing these skills into a single transferable skill and sub-goal.” However, in this work a finite number of options is needed and the result is a potentially infinite number of options, not a single one. Similarly, it is mentioned that “Touati et al. (2022) assumes access to a distribution of relevant tasks”, but this work introduces a reward-agnostic approach, so no distribution of relevant tasks is needed.
> > >
> > > Suggestions:
> > >
> > > Considering the authors’ reply and the revised version, I think the paper fails by focusing its narrative and motivation in the pitfalls of works that are related but that are too different in the way they work to be comparable, and that in addition might not be as problematic as implied. I agree that it does not make sense to compare with methods that are too different, but at the same time I do not agree with the paper being so novel that a single proof of work is sufficient to get an acceptance. To me, the paper should include some experimental evidence that the ensemble method being proposed in combination with any subgoal discovery method provides any type of advantage over 1) using an appropriate online subgoal state discovery method in combination with some HRL method, and 2) any other online technique that deals with spurious correlations, also in combination with some HRL method.

---

> > > > ### Author Response · Authors · 2024-11-25
> > > >
> > > > We thank the reviewer for their response and willingness to discuss.
> > > >
> > > > The benefit of using an ensemble is that if you are learning to transfer a subgoal from a single example, it is not obvious what features are relevant and what are not. Using the example from the paper, if we have a subgoal that is defined over an open cupboard which contains a red ball, is the subgoal opening the cupboard or finding the red ball? There are tasks where both of these hypotheses would be beneficial. If we wish to make coffee and the mug is inside a closed cupboard, opening the cupboard door is more useful than finding a red ball - which may not be in the new cupboard. If we wish to place a red ball inside a container, identifying a red ball, which may no longer be inside a cupboard is the most useful subgoal. While only one of these subgoals may be relevant to a single task, that is not to say that only one of the subgoals is ever useful, it just depends on the task that is defined. By learning an ensemble we can break a single subgoal, which will consist of several confounding factors, into a collection of subgoals that separate these factors for use in future tasks.
> > > >
> > > > While not all skill discovery methods are subgoal based, a common method of skill discovery is to identify a collection of states that are interesting in some way and learn a policy to hit that set of states.
> > > >
> > > > In our experiments we do not perfectly collect all states that encompass our subgoals in a single task. In the Montezuma'sRevenge experiment, we see that the states collected show little difference in performance after two ladders. This is the same data we used to train the ensembles for the policy experiment and so does not perfectly capture all the relevant states. For our key data in MiniGrid, all data collected has a door colour that corresponds to the key colour, so blue key data has a blue door. This skill still correctly collects the blue key but this is irrelevant to the task, which only requires a red key and so none of these ensemble members are selected by the high-level policy.
> > > >
> > > > By generalizing a previously identified subgoal, this allows us to group options that we learn in the future as we learn them. This results in a distribution of tasks, all linked to a single, transferred subgoal, that would allow us to, for example, sample from to learn a general policy that could be used in future. Transferring subgoals also allows for the creation of smaller skill libraries that can be accessed by a general agent instead of continually learning new skills. The ensembles allow us to turn a single subgoal into a set of subgoals, each with fewer confounding factors.
> > > >
> > > > We will correct the relevant works. However, we will point out that Touati et al. (2022) is designed to transfer to changes in the reward function, while we wish to transfer to unseen states. Barreto et al. (2019) combines previously learnt options that are combined to create a new option. These new options are combinations of the previously learnt options and so is more analogous to an idea like skill-chaining.
> > > >
> > > > We have made updates to the introduction and related works.
> > > >
> > > > Thank you for engaging us in this discussion.

---

### Official Review · Reviewer_M7Hj · 2024-11-04

**Soundness:** 2
**Presentation:** 3
**Contribution:** 3
**Rating:** 3
**Confidence:** 3

**Summary:**

The paper presents a new method for skill discovery in hierarchical reinforcement learning which is designed to find skills which can be effectively transferred between tasks.
In order to do this, the authors focus on learning sub-goals with representations that facilitate this transfer by training ensembles of diverse classifiers for use as sub-task termination predicates.
The authors detail several experiments which evaluate different aspects of their method.

**Strengths:**

The paper addresses the important topic of learning transferrable skills for hierarchical RL and is generally well-written.

The method presented is a novel and interesting application of a diversity-based classification method to skill discovery.

Several experiments are included which are each intended to answer a different useful question.

**Weaknesses:**

Some parts of the experiments are unclear and overall the presented results are not convincing. Specific points follow.

For the quantitative results (Figures 3, 4, 5), I don't think that three seeds per curve is sufficient to reasonably compare the methods. While there isn't a definitive way to determine how many are needed, I would expect to see at least five to assess the variance due to random model initialization when the environment is deterministic.

In section 4.1, it is not clear what the ensembles are being compared against to determine the accuracy. I'm also not sure I agree with the claim that the performance when only one ladder instance has been seen is the most important given how low the accuracy is for all three models at that point compared to the two-ladder case.

For the MiniGrid results in Section 4.3, I am concerned about the lack of success with DQN and PPO. This environment is not so large that DQN/PPO should be unable to succeed when trained over 1.5 million steps, so it makes me think there is an error in the evaluation. Based on the hyperparameters in the appendix, one potential major factor for DQN is a lack of exploration.

The training process in Section 4.3 is not clear. It seems to state that data is collected from the different seed runs and used together to train the classifier ensembles, but this doesn't make sense with results being aggregated across the three seeds.

**Questions:**

As mentioned above, can you explain the baseline how the accuracy is calculated in section 4.1?

As mentioned above, can you better explain the training process for the MiniGrid environment in section 4.3?

In Figure 6, why is the standard error shown instead of standard deviation?

$ $

Minor Notes:

Line 320: I believe this is meant to be a reference to Figure 4 rather than Figure 5.

Manhattan is misspelled several times.

---

> ### Author Response · Authors · 2024-11-15
>
> Thank you for your thoughtful review. We agree that our presentation can be improved. In particular, we should first clarify that our work is not a new method for skill discovery. It is a method to generalize subgoals and so should be used in tandem with a skill discovery method.
> > I don't think that three seeds per curve is sufficient to reasonably compare the methods
>
> We are actively running more seeds and will update all graphs with additional seeds.
>
> > can you explain the baseline how the accuracy is calculated in section 4.1?
>
> For evaluation, we collected data from across Montezuma’sRevenge and labeled which states lie within and outside the subgoal. We calculate our accuracy against these expert labeled samples.
>
> > I'm also not sure I agree with the claim that the performance when only one ladder instance has been seen is the most important given how low the accuracy is for all three models at that point compared to the two-ladder case.
>
> The goal of our work is to generalize a subgoal from a single example, which is the expected output of an off-the-shelf subgoal discovery algorithm. As such, illustrating how well our method performs for the one-ladder case indicates how well the method works if used along-side a subgoal discovery method. Although 70% accuracy may appear to be low at first glance, especially when compared to the near 100% accuracy achieved after two input examples, it means that we detect previously unseen states that fall inside our ClimbDownLadder subgoal—each detection means a new skill need not be discovered from scratch, but can simply be instantiated as a generalization.
>
> > This environment is not so large that DQN/PPO should be unable to succeed when trained over 1.5 million steps, so it makes me think there is an error in the evaluation.
>
> The failure of the baselines is due to the episode length of 500 steps, which makes it a more challenging exploration problem. If we increase the episode length to 1000 steps, both DQN and PPO solve the task. We chose these baselines because our hierarchical agent uses  DQN for its lower level policies and PPO is used for the higher-level policy. Our intention is to convincingly show that our agent learned the task not because of the power of these deep RL methods but because of the subgoals provided by our ensembles. We will clarify this argument in the revision to our paper.
>
> > can you better explain the training process for the MiniGrid environment in section 4.3?
>
> We collect labeled data from a single seed (seed=0) and unlabeled data is collected from seeds 1 and 2. We then evaluate the agent in unseen seeds (10, 11, 12). When training starts, the ensembles have seen only data from seeds 0, 1 and 2. The additional states seen in the testing seeds are not shared across the ensemble agents. This was not clearly communicated in the current paper.  We can address this completely with a careful experiment description, in revisions we will implement.
>
> > In Figure 6, why is the standard error shown instead of standard deviation?
>
> Standard error is standard deviation divided by the square root of the number of runs; this is a standard metric, but we can report standard deviation if you think that will substantially increase the quality of the paper.
>
> Thank you again for your thoughtful review. The points you raise will be reflected in our revisions and we welcome any additional questions.

---

> > ### Author Response · Authors · 2024-11-17
> >
> > Dear reviewer,
> >
> > We have updated our paper draft with the following changes
> >  - Clarified our goal in the introduction
> >  - Added additional related works (mostly to the Learning generalized skills section)
> >  - Clarified the algorithm methodology, adding mathematical context for hypotheses and improving descriptions of how D-BAT fits into the algorithm
> >  - Clarified experimental set-up and decisions
> >
> > We will be updating our experiments with additional seeds once all runs have completed.
> >
> > Please let us know if you have any further questions or concerns.

---

> ### Author Response · Authors · 2024-11-21
>
> Dear reviewer,
>
> We have updated all experiments to be averaged over 10 seeds.
>
> We believe we have addressed all your concerns and are eager to engage with you on any further points or questions you may have!

---

> > ### Comment · Reviewer_M7Hj · 2024-12-03
> >
> > I appreciate the responses and updates made by the authors.
> >
> > It is good that you have added more runs to increase the number of seeds per algorithm. The new results seem to show little benefit to the D-BAT Ensemble over the Standard Ensemble at almost all points in Figures 3 and 4. This is much weaker support for your method.
> >
> > DQN and PPO are not unreasonable algorithms to compare against, but I remain concerned about their lack of success in this environment. Figure 6 seems to indicate that not one of the runs for DQN or PPO succeeded in learning a good policy. While I understand different step limits for truncation could change things, it is not a convincing comparison to use a version of the environment for which these algorithms completely fail. If DQN and PPO can succeed with 1000-step truncation, then it makes much more sense to use that version.
> >
> > It is possible that I am still missing something, but you shouldn't be mixing data from different seeds. The purpose of doing runs with different seeds is to allow some quantification of how the algorithm is affected by random parts of the algorithm governed by the seed, e.g., model initialization, exploration, etc.
> >
> > Standard error and standard deviation quantify different things. SE has to do with how accurately the mean has been estimated while SD has to do with the spread of the data. A reason that I asked was that the other graphs all reported standard deviation. Is Figure 6 meant to show something different?

---

### Official Review · Reviewer_grQD · 2024-11-04

**Soundness:** 2
**Presentation:** 1
**Contribution:** 2
**Rating:** 3
**Confidence:** 4

**Summary:**

Learn a collection of hypotheses which take in varying collections of features using D-BAT, a method which . Then identify the hypothesis that has the best transferability. Relies on labeled training data for subgoals. Then give the agent the transferable subgoals.

**Strengths:**

The problem of robust skill generalization is necessary.

The motivation for the problem is clear

**Weaknesses:**

This work makes several overgeneralizing claims about the positioning of HRL. Some examples include (line 36) that the options based framework utilizes subgoals (the termination condition does not need to be goal based). The termination condition must be a set (line 80), when it is often formulated as a probabilistic condition function. The equivalence between termination sets and subgoals (line 83), since a set might contain many subgoals. The use D-BAT to claim generalizability (line 162): While there may be some amount of robustness added, there is an implicit assumption that by learning a robust classifier of subgoals, this implies that the same assumptions can be applied to goals in RL, which is both a different context and not necessarily true. This point would need to be proven theoretically and the empirical results would need to support this claim more directly.

The most glaring weakness is the level of imprecision realted to the method itself. In particular, it is not clear what the algorithm actually is. It appears to be 1) run D-BAT to get some features. 2) learn an option to achieve good hypothesis classification. However, it is not made clear what the inputs are for D-BAT, the reward function for the skills, the hierarchy, or almost any other detail of the algorithm.

The experimental results lack several components. First, the baselines compared are deep RL algorithms, not state of the art HRL algorithms. Second, neither method uses factorization or exploration. Third, the main paper lacks even a complete coverage of the tasks, since downstream performance is only evaluated in one task.

Finally, the work is entirely not self contained, since it relies heavily on D-BAT, without actually providing an adaquate formalism to describe D-BAT. Instead, the reader is expected to read this work. Furthermore, it is not obvious why this particular hypothesis algorithm is chosen over others, nor are there any ablations to indicate that D-BAT is preferable to other robust hypothesis algorithms---only comparison to a CNN.

**Questions:**

See Weaknesses

---

> ### Author Response · Authors · 2024-11-15
>
> Thank you for  your comments which expose a weakness in our presentation which we will address. Specifically, the goal of the paper could be misconstrued as applying D-BAT to the hierarchical reinforcement learning setting. This would indeed limit the novelty significantly to the D-BAT method which would not justify publication. We want to emphasize that this is not the main goal of our work. Current subgoal discovery methods (which are the vast majority of skill discovery methods) identify subgoals which are considered standalone and unique. Current skill generalization methods take a retroactive approach, by either assuming a large library of learnt skills which are then compressed into a small collection of generalized skills or assuming the agent has access to a distribution of tasks to sample from during training. This latter case implicitly assumes an expert has informed the agent of what skills it must have. This retroactive view means current skill generalization techniques are not compatible with the rich collection of skill discovery algorithms the community has developed.
>
> We take a forward looking approach, generalizing a unique subgoal, which can be provided by any subgoal-based skill discovery method, to unseen states. This leaves two main problems 1) we have a subgoal described using  a small labeled dataset, but with very low diversity/coverage, and 2) based purely on one instance, the true intention of the subgoal is unclear (we argue this with an illustrating example in section 3.2). To overcome these challenges, we propose learning an ensemble of classifiers, each achieving high accuracy on the training dataset, but attending to separate features of the data. We make use of D-BAT because it is the only method we found capable of creating multiple classifiers that vary meaningfully on out-of-distribution data with their use of unlabeled data. So, while D-BAT can be regarded as a robustness algorithm, we are not using it to increase robustness of a provided subgoal, but instead to create a collection of subgoals based on a sufficiently descriptive subset of the data. As such, our method is not tied to the D-BAT method and any algorithm that produces a set of diverse classifiers can replace D-BAT.
>
> Each experiment was carefully selected. For our subgoal classifier evaluation, we compare to a CNN because this is traditionally how subgoals are represented in the vast majority of GCRL and HRL works.  In the MiniGrid-DoorKey experiment, we show two things: (1) giving the agent skills that are not aligned with the overall task will not irreparably damage policy learning, and (2) the strong performance of our ensemble agent cannot be explained by our use of DQN (for learning the policy over options) and PPO (for learning low-level option policies).
>
> As the review report correctly indicates, this is not communicated clearly in the paper. We did not compare to state-of-the-art skill discovery methods because our methodology can be applied with any subgoal based skill discovery method. We did not compare against any skill generalization techniques because these methods either assume prior knowledge of the collection of tasks the skill must generalize to or they assume a large library of prior learnt skills. These would therefore not be a fair comparison.
>
> We will improve the current draft of the paper by a) removing the overgeneralizations of HRL b) better communicating the goal of our work and our algorithm c) better explaining the selection of comparisons in our experiments.
>
> Thank you again for a thoughtful review, we are happy to address any additional concerns.

---

> > ### Author Response · Authors · 2024-11-17
> >
> > Dear reviewer,
> >
> > We have updated our paper draft with the following changes
> >  - Clarified our goal in the introduction and removed overgeneralizations of HRL
> >  - Added additional related works (mostly to the Learning generalized skills section)
> >  - Clarified the algorithm methodology, adding mathematical context for hypotheses and improving descriptions of how D-BAT fits into the algorithm
> >  - Clarified experimental set-up and decisions
> >
> > We will be updating our experiments with additional seeds once all runs have completed.
> >
> > Please let us know if you have any further questions or concerns.

---

> ### Author Response · Authors · 2024-11-21
>
> Dear reviewer,
>
> We have updated all experiments to be averaged over 10 seeds.
>
> We believe we have addressed all your concerns and are eager to engage with you on any further points or questions you may have!

---

> ### Comment · Reviewer_grQD · 2024-11-25
> **Response to Authors**
>
> Looking over the updated paper, while the addition of seeds is important, the limitations remain. In particular, there is still an absence of rigor related to the hypothesis formulation and how it is precisely applied to HRL. It is still entirely unclear because of this lack of formalism what the inputs or outputs are, and these must be interpolated from the related work, rather than from the paper itself. Speaking of this, the paper continues to not be self-contained, without a formal description of D-BAT prior to it's inclusion in the methods. Next, the notable absence of competitive HRL baselines (The argument that these methods are not directly comparable does not mean that they cannot be compared, and further suggests that a comparable baseline should be used. Furthermore, if the claim is that this method is strictly tangential, then experiments should be run to indicate that it can actually augment existing methods, and a comparison against the un-augmented version). Finally, the edits to the paper are not sufficiently notable to improve the quality of the paper---it remains difficult to parse.
>
> For these reasons I will maintain that the paper should be rejected.

---

> > ### Author Response · Authors · 2024-11-26
> >
> > Thank you for your reply.
> >
> > We have made minor updates to the related works and method section.

---

### Meta-Review · Area_Chair_ygK5 · 2024-12-20

**Metareview:**

This paper proposes a new method for learning transferable skills in hierarchical reinforcement learning (HRL) by focusing on the generalization of subgoals. The key idea is to use an ensemble of diverse classifiers, trained with a technique called D-BAT, to represent different hypotheses about which state features are important for achieving a subgoal.

Strengths
-----------
- **Novelty:** The paper introduces a new way to think about skill transfer in HRL, emphasizing the role of subgoal generalization and leveraging the D-BAT algorithm to learn diverse representations.
- **Promising results:** Experiments in MiniGrid and Montezuma's Revenge suggest that the proposed method can enable agents to learn transferable skills and solve challenging tasks.

Weaknesses
-------------
- **Lack of clarity:** Several reviewers criticized the paper's presentation, finding it imprecise and difficult to understand the specific details of the algorithm. Key aspects like the inputs to D-BAT, the reward function for skills, and the overall hierarchy were not clearly explained.

- **Limited evaluation:** The experimental results were considered insufficient, with concerns about the choice of baselines, the lack of comparison to state-of-the-art HRL methods, and limited evaluation of downstream performance.

- Overgeneralization:** The paper makes some broad claims about HRL that may not be entirely accurate, and the connection between robust subgoal classification and transferable skills in RL needs further justification.

- **Dependence on D-BAT:** The paper relies heavily on the D-BAT algorithm without providing a sufficient explanation, making it difficult for readers to understand the method without consulting external sources.

This paper presents an intriguing idea for learning transferable skills in HRL, but the majority of reviewers recognize that it suffers from a lack of clarity and limited evaluation that make it not ready for publication. I recommend the authors addressing reviewers' concerns for future improvement.

**Additional Comments On Reviewer Discussion:**

The discussion did not change reviewers' concerns on the limitations of the empirical analysis.

---

### Decision · Program_Chairs · 2025-01-22

Reject